# Predictors of Response and Survival in Immune Checkpoint Inhibitor-Treated Unresectable Hepatocellular Carcinoma

**DOI:** 10.3390/cancers12010182

**Published:** 2020-01-11

**Authors:** Pei-Chang Lee, Yee Chao, Ming-Huang Chen, Keng-Hsin Lan, Chieh-Ju Lee, I-Cheng Lee, San-Chi Chen, Ming-Chih Hou, Yi-Hsiang Huang

**Affiliations:** 1Institute of Pharmacology, School of Medicine, National Yang-Ming University, Taipei 11221, Taiwan; tympanum3688@gmail.com (P.-C.L.); khlan@vghtpe.gov.tw (K.-H.L.); 2Faculty of Medicine, School of Medicine, National Yang-Ming University, Taipei 11221, Taiwan; iclee@vghtpe.gov.tw (I.-C.L.); scchen16@vghtpe.gov.tw (S.-C.C.); mchou@vghtpe.gov.tw (M.-C.H.); 3Division of Gastroenterology and Hepatology, Department of Medicine, Taipei Veterans General Hospital, Taipei 11217, Taiwan; ssbugi@gmail.com; 4Department of Oncology, Taipei Veterans General Hospital, Taipei 11217, Taiwan; mhchen9@vghtpe.gov.tw; 5Institute of Clinical medicine, School of Medicine, National Yang-Ming University, Taipei 11221, Taiwan

**Keywords:** alpha fetoprotein response, immune checkpoint inhibitor, unresectable hepatocellular carcinoma

## Abstract

Immune checkpoint inhibitors (ICIs) with nivolumab and pembrolizumab are promising agents for advanced hepatocellular carcinoma (HCC) but lack of effective biomarkers. We aimed to investigate the potential predictors of response and factors associated with overall survival (OS) for ICI treatment in unresectable HCC patients. Ninety-five patients who received nivolumab or pembrolizumab for unresectable HCC were enrolled for analyses. Radiologic evaluation was based on RECIST v1.1. Factors associated with outcomes were analyzed. Of 90 patients with evaluable images, the objective response rate (ORR) was 24.4%. Patients at Child–Pugh A or received combination treatment had higher ORR. Early alpha-fetoprotein (AFP) >10% reduction (within 4 weeks) was the only independent predictor of best objective response (odds ratio: 7.259, *p* = 0.001). For patients with baseline AFP ≥10 ng/mL, significantly higher ORR (63.6% vs. 10.2%, *p* < 0.001) and disease control rate (81.8% vs. 14.3%, *p* < 0.001) were observed in those with early AFP reduction than those without. In addition, early AFP reduction and albumin-bilirubin (ALBI) grade or Child–Pugh class were independent factors associated with OS in different models. In conclusion, a 10-10 rule of early AFP response can predict objective response and survival to ICI treatment in unresectable HCC. ALBI grade and Child–Pugh class determines survival by ICI treatment.

## 1. Introduction

Hepatocellular carcinoma (HCC) is the fifth most common cancer and the second leading cause of cancer related death worldwide that constitutes a major global health problem [1,2]. Despite improvement in surveillance and hepatitis B vaccination, hepatitis C treatment, a large number of patients still present with unresectable, advanced-stage disease and require systemic therapy [2]. Sorafenib has long been the first and the only effective systemic treatment for advanced HCC [3,4]. Recently, several positive results from the phase 2/3 trials of first or second line settings enable HCC patients access to more treatment options [5,6,7].

Manipulation of immune checkpoints by targeted antibodies, such as anti-programmed cell death-1 (PD-1) antibody, has recently emerged as an effective anticancer strategy for many types of cancers including HCC [8]. Nivolumab and pembrolizumab, the anti-PD-1 antibodies, are FDA conditionally approved immune checkpoint inhibitors (ICIs) for HCC as a second line treatment after sorafenib failure [5,6,9]. Based on the multi-cohort phase 1/2 trial CheckMate-040, phase 2 trial keynote-224, and phase 3 trial of Keynote-240 [5,6,7], only 14–18% of HCC patients could get a tumor response by nivolumab or pembrolizumab. Traditionally, PD-L1 expression level is a determinant marker of response in lung cancer, gastric cancer, head and neck cancer, and urothelial cancer [10,11,12]. However, previous CheckMate-040 and Keynote-224 studies could not show a significant association between PD-L1 expression level and tumor response in HCC [5,6]. As ICI treatment is expensive and has potential risk of immune-related adverse events, a baseline or early biomarker can help physicians to encourage suitable patients to maintain the treatment [2]. However, so far, it still remains an unmet medical need as there is no well-identified biomarker for HCC immunotherapy. In this study, we aimed to identify potential predictors of treatment response and overall survival (OS) in patients treated with ICI for unresectable HCC.

## 2. Results

### 2.1. Demographic Characteristics of the Study Cohort

Upon enrollment, most patients were within Child–Pugh class A (72.6%); but more than half of them were classified beyond albumin-bilirubin (ALBI) grade 1 (71.6%). A total of 78.9% of the patients were at BCLC stage C, and the maximal tumor size was 5.2 cm (IQR, 2.3–8.8). The median alpha-fetoprotein (AFP) level was 865.6 ng/mL, and 15.8% of the patients had low AFP level (<10 ng/mL). In addition, 41.1% received ICI as first-line systemic therapy, while 58.9% had experienced sorafenib failure. Among 95 patients, 13 received combination therapy with ICIs and tyrosine kinase inhibitors (six with sorafenib, six with lenvatinib, and one with regorafenib). Four and three patients developed grade 2 immunotherapy-related pneumonitis and hepatitis, respectively. Six patients suffered from grade 1/2 skin reactions. The detailed baseline characteristics are presented in Table 1.

### 2.2. Treatment Response to ICI Therapy

The median duration of ICI treatment was 10.4 weeks (IQR, 4.8–22.3) with a median of five cycles (ranged 1–35) administered. As presented in Table 2, the disease control rate (DCR) was 36.7%, including six complete response (CR), 16 partial responses (PR), and 11 stable diseases. The best objective response rate (ORR) was 26.9% and 20.0% between patients at Child–Pugh A and B, respectively. Combination treatment had a significantly higher ORR than ICI monotherapy (46.2% vs. 20.8%, *p* = 0.049). The median time to response was 63 days (IQR, 48–75) after a median five cycles of ICI treatment (IQR, 4–6); and the median duration of response was not yet reached for responders (16/22 kept ongoing with response). Noteworthily, three Child–Pugh B patients whose tumors controlled well by ICI notably improved their liver reserve to Child–Pugh A after treatment.

In univariate analysis, AFP >10% reduction within the first 4 weeks of treatment, baseline ALT level, as well as combination treatment were associated with best objective response. In multivariate analysis, early AFP response was the only independent predictor of best objective response to ICI treatment (odds ratio: 7.259, *p* = 0.001) (Table 3). Besides, early AFP reduction was also associated with best disease control by ICI therapy (Appendix A).

### 2.3. Association between Tumor Response and Early AFP Response

As 10% AFP reduction might not be meaningful for HCCs with baseline level less than 10 ng/mL, the AFP response was further categorized by baseline AFP level. For patients with baseline AFP ≥10 ng/mL, significantly higher ORR (63.6% vs. 10.2%, *p* < 0.001) and DCR (81.8% vs. 14.3%, *p* < 0.001) were observed in those with early AFP reduction than those without. However, such association was not observed in patients with baseline AFP level <10 ng/mL (Figure 1).

### 2.4. Response in HCC Patients with Available PD-L1 Level and Evaluable Images

Of 18 patients whose tumor specimens were assessed for PD-L1 expression, three patients had TPS ≥ 1% or CPS ≥ 1%, and all of them achieved partial response to ICI treatment. In the other 15 patients with low expression of PD-L1 (<1%), 60.0% developed PD (*p* = 0.206) (Appendix A).

### 2.5. Uni- and Multivariate Analysis for Factors Associated with OS for All HCC Patients

During a median follow-up period of 5.2 (IQR, 3.2–12.5) months, 47 deaths occurred. The median overall survival was 11.9 months (95% C.I. 5.6–18.2). As shown in Figure 2, patients with objective tumor response had significantly better OS than those that developed PD (median OS: not yet reached vs. 6.1 months). Besides, patients with early AFP reduction >10% also had significantly better OS than non-responders (median OS: 24.7 vs. 5.6 months, *p* = 0.014; Figure 3). In addition, Child–Pugh A vs. B/C (median OS: 24.7 vs. 3.8 and 0.6 months, *p* < 0.001) (Figure 4A), and ALBI grade 1 vs. 2/3 (median OS: not yet reached, vs. 5.6 and 3.2 months; *p* < 0.001) (Figure 4B) were associated with OS. No significant survival difference was reported according to prior sorafenib treatment (Appendix A).

As declared in Table 4, early AFP reduction (hazard ratio (HR): 0.234, *p* = 0.001) and Child–Pugh A (HR: 0.238, *p* = 0.002) were the independent predictors to better OS in patients received ICI treatment (Multivariate analysis model 1). Similarly, early AFP response (HR: 0.243, *p* = 0.001) and ALBI grade 1 (HR: 0.220, *p* = 0.002) were also good survival predictors in the model 2. After including tumor response into analysis, presence of tumor response, serum AST level, and good liver reserves were identified as independent survival predictors (Appendix A).

## 3. Discussion

This is the largest real-world cohort from Asian patients with unresectable HCC treated by ICIs until now. A better ORR (24.4%) was observed than previous studies, but stable disease was fewer [7,8,13]. Impressively, early AFP response within 4 weeks of treatment was identified as the independent predictor to objective response in our patients. Besides, better liver reserves (Child–Pugh class A or ALBI grade 1) and early AFP response were also good predictors of survival.

The predictive role of AFP reduction in HCC response to various treatments has been reported [14,15,16]. In sorafenib-treated HCC, a decline of AFP >20% from baseline level after 4 to 8 weeks of treatment was suggested as a surrogate marker to predict treatment response and survival benefits [17]. In an extended analysis of CheckMate-040, however, the authors failed to find biomarkers predicting treatment response to nivolumab [18]. In a recent real-life experience of ICI-treated HCC, no factor was identified to associate with response, either [19]. Early AFP reduction >20% within the first 4 weeks of ICI treatment was recently reported in relation to treatment efficacy for patients with baseline AFP > 20 ng/mL [20]. In this study, we proposed a novel 10-10 rule to early predict ICI response based on baseline AFP level ≥10 ng/mL, and 10% reduction within 4 weeks of treatment. A >10% reduction of AFP (ORR: 63.6% vs. 10.2%, *p* < 0.001; DCR: 81.8% vs. 14.3%, *p* < 0.001) performed a better discriminative ability in tumor response than AFP reduction >20% (ORR: 64.7% vs. 14.8%, *p* < 0.001; DCR: 82.4% vs. 20.4%, *p* < 0.001) or >30% (ORR: 61.5% vs. 19.0%, *p* = 0.001; DCR: 84.6% vs. 24.1%, *p* < 0.001). These findings suggested the 10-10 rule can serve as guidance for ICI treatment in advanced HCC.

Unlike the real-world report of sorafenib-treated HCC with inferior survival benefit in Eastern population [21], the median OS of our patients was similar to the data of CheckMate-040 and a recent multicenter real-world study [7,19]. Our results were also in line with the report from Asian cohort of CheckMate-040 with comparable ORR and survival [22]. The prognosis of advanced HCC depends not only on tumor burden, but also on liver reserve [21,23,24]. Consistent with the survival-predictive ability of ALBI grade in sorafenib-failed HCC [21,25], our data confirmed ALBI grade as an independent survival predictor in patients received ICI treatment. The ORR was 12.2% for Child–Pugh B HCC in CheckMate-040 [26]. In a recent case series enrolled 18 patients with advanced HCC and Child–Pugh B cirrhosis, the ORR was 17%, and the median OS was 5.9 months [27]. In this study, we declared a comparable ORR as 20.0% (three PR and one CR) in Child-Pugh B patients but presented with a shorter OS. Although only five of our Child–Pugh B patients had disease controlled by ICI treatment; notably, three of them improved their liver function to Child–Pugh A along with excellent survival benefits (the median OS was not yet reached). Inconsistent with prior statements indicated that patients with poor liver reserves may not get benefit from oncological management [13,28], these recent findings suggested that Child–Pugh B patients could still get benefit from ICI treatment.

Synergic benefits of combination therapy to advance-staged HCC have been explored recently [29]. Current ongoing clinical trials suggested that combination treatment with lenvatinib plus pembrolizumab, atezolizumab plus bevacizumab, or nivolumab plus ipilimumab had promising ORR higher than 30%, or even 60% [30,31,32]. As recently reported in ESMO Asia 2019, the phase 3 IMbrave 150 has demonstrated significant improvements with atezolizumab and bevacizumab over sorafenib in OS and RFS for unresectable HCC (ORR 27%, DCR 74% by RECIST 1.1). In this study, a significantly better ORR (46.2%) was noted in patients that received combination therapy compared with ICI monotherapy. However, it did not independently predict objective response to ICI treatment in the multivariate analysis. Further investigation is still needed to clarify the role of combined treatment in management of HCC.

PD-L1 expression by either tumor cells or intratumoral inflammatory cells is related to HCC aggressiveness and might account for the response of immune checkpoint inhibitor [33]. Although numerically a higher ORR was observed in patients whose PD-L1 expression level was ≥1% in previous studies [7,8], the difference did not reach statistical difference. In our data, all patients with ≥1% TPS or CPS had PR to ICI treatment; whereas, most patients with <1% PD-L1 expression presented with PD. These findings suggested that PD-L1 expressions might play some role in the response to ICI treatment in HCC.

The adverse effects (AEs) of immune checkpoint inhibitor are different from toxicities caused by chemotherapy. In general, the immunotherapy related AEs was low in our cohort and we did not observe a high incidence of immunotherapy related AEs in Child–Pugh B patients.

There are several limitations in this study. First, this is a retrospective study that only enrolled patients in single hospital. However, our hospital is the main leading tertiary medical center in Taiwan. The information bias would be ameliorated by regular tumor reassessment by contrast-enhanced image and clinical evaluation. Besides, it is so far the largest real-life Asian ICI-treated HCC cohort; and is the first study demonstrated the 10-10 role of AFP to predict response. Second, the level of PD-L1 expression was only performed in few patients, although our pivotal results were similar to previous studies with improved prediction to treatment efficacy [7,8]. Third, most of our patients (73.5%) had chronic hepatitis B as the underlying hepatic disease. Our results should be applied to other populations with caution.

## 4. Materials and Methods

### 4.1. Patients

From May 2017 to August 2019, 95 patients had received nivolumab or pembrolizumab treatment for unresectable HCC in Taipei Veterans General Hospital and were retrospectively enrolled in this study. None of them enrolled in previous or ongoing ICI clinical trials. Among them, 90 patients with evaluable image studies following the treatment before the cut-off date of data were recruited for further assessment of treatment response. Of the five subjects not available for assessment, four patients died before the first radiological evaluation and one patient was lost to follow-up. The diagnosis was according to the AALSD treatment guidelines for HCC [34]. ICIs were prescribed to these patients because of treatment failure or intolerable adverse events to sorafenib, deteriorated liver reserves beyond Child-Pugh class A so that was unable to apply sorafenib according to the reimbursement criteria of National Health Insurance in Taiwan [21], or patients who experienced ineffective transarterial chemoembolization for their intermediate-staged HCC. The study was approved by the Institutional Review Board of the Taipei Veterans General Hospital (IRB numbers: 2017-09-007CC, 2019-07-007AC, and 2019-08-006B). All alive patients have signed informed consent; and the informed consent of others was waived by IRB because of retrospective design.

### 4.2. Treatment and Outcome Assessment

ICIs were administered according to the recommended dosing and safety information (2–3 mg/kg, every 2 weeks for nivolumab and every 3 weeks for pembrolizumab). The safety assessment and grading was performed based on the National Cancer Institute Common Terminology Criteria for Adverse Events (NCI CTCAE; version 4.03). Besides, clinical evaluation with Child-Pugh class, albumin-bilirubin grade [35,36], hemogram, serum chemistry, and alpha-fetoprotein (AFP) level were performed every 2 to 3 weeks during the treatment. An early AFP response was defined as >10% reduction from baseline level within 4 weeks of treatment. 

Clinical tumor response was assessed by RECIST version 1.1 based on contrast-enhanced abdominal computed tomography scan or magnetic resonance imaging [7,37]. The image examinations were carried out every 6–8 weeks during ICIs treatment. The OS was measured from the date of starting ICIs until the date of death; and the time to response was the interval between ICIs initiation and occurrence of first objective response. 

### 4.3. PD-L1 Expression Analysis

PD-L1 expression was measured by immunohistochemistry pharmDx assay (Agilent Technologies, Santa Clara, CA, USA) on archive HCC tissues for 18 patients. The anti-PD-L1 28-8 antibody was used for nivolumab-treated HCC, and anti-PD-L1 22C3 antibody was applied for pembrolizumab-treated HCC [7,8]. Expression levels were reported by tumor proportion score (TPS) and/or combined positive score (CPS), respectively [7,8].

### 4.4. Biochemical Tests

Serum biochemistry tests were measured by systemic multi-autoanalyzer (Technicon SMAC, Technicon Instruments Corp., Tarrytown, NY, USA). Serum AFP levels were measured by chemiluminescent microparticle immunoassay (ARCHITECT AFP assay, Abbott Ireland Diagnostics Division, Sligo, Ireland) with clinically reportable range from 1 to 1,998,000 ng/mL.

### 4.5. Statistical Analysis

Continuous variables were expressed as median (interquartile ranges—IQR), while categorical variables were analyzed as frequency and percentages. The Pearson chi-square analysis or Fisher’s exact test was used to compare categorical variables, while the Student’s *t*-test or Mann–Whitney U test was applied for continuous variables. Survival was estimated by the Kaplan–Meier method and compared by the log-rank test. Additionally, Cox’s proportional-hazard model was used to identify prognostic factors for survivals. To avoid the effect of collinearity, ALBI grade and BCLC or Child-Pugh class were not included in the same multivariate model. For all analyses, *p* < 0.05 was considered statistically significant. All statistical analyses were performed using the Statistical Package for Social Sciences (SPSS 17.0 for Windows, SPSS Inc., Chicago, IL, USA).

## 5. Conclusions

The 10-10 rule of early AFP response can predict objective response and survival to ICI treatment in unresectable HCC. Besides, good liver reserves confer better survival among these patients. These findings help to provide effective on-treatment guidance of ICI treatment for HCC patients.

## Figures and Tables

**Figure 1 cancers-12-00182-f001:**
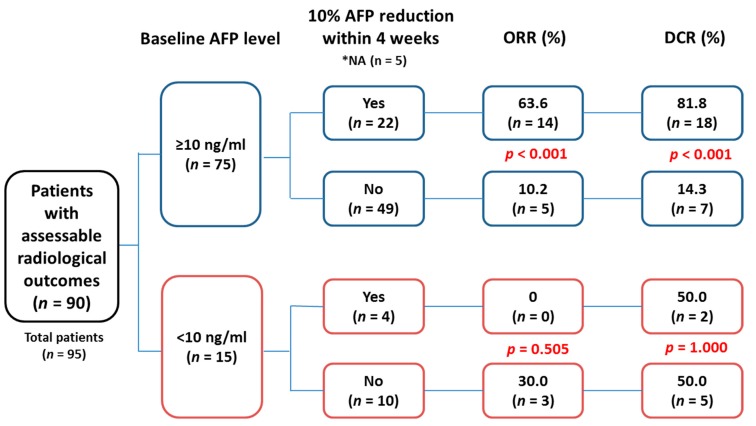
The association between tumor response and early alpha-fetoprotein (AFP) reduction categorized by AFP ≥ or <10 ng/mL. NA (not assessed): total of five patients did not have an AFP value within 4 weeks of treatment that could not be assessed for early AFP response.

**Figure 2 cancers-12-00182-f002:**
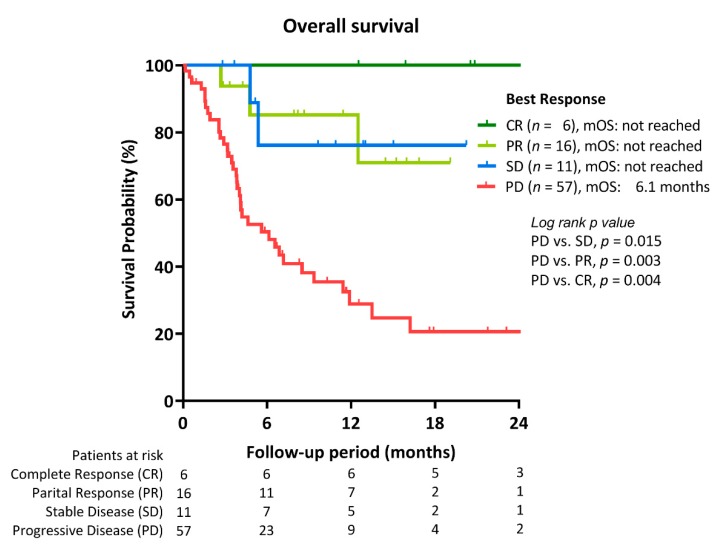
Overall survival (OS) of unresectable HCC according to treatment response to immune checkpoint inhibitors.

**Figure 3 cancers-12-00182-f003:**
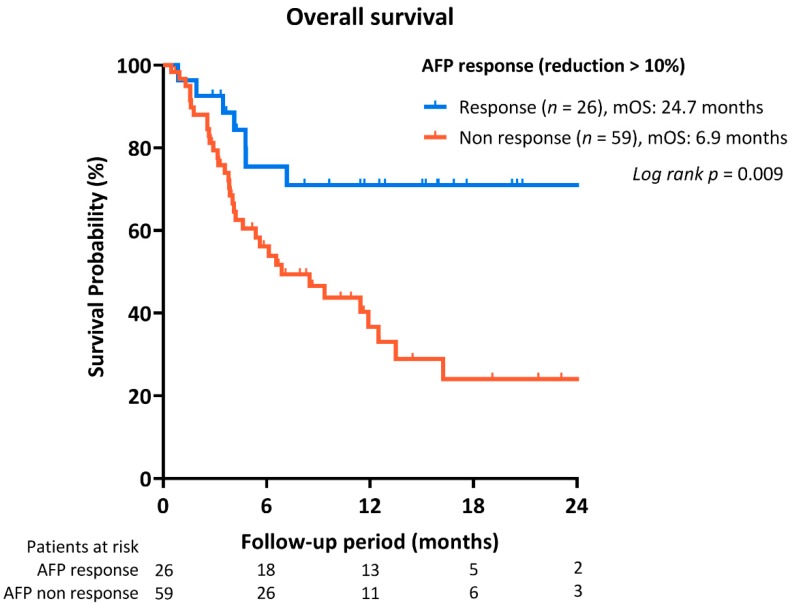
OS of HCC patients according to AFP reduction within 4 weeks treatment of immune checkpoint inhibitors.

**Figure 4 cancers-12-00182-f004:**
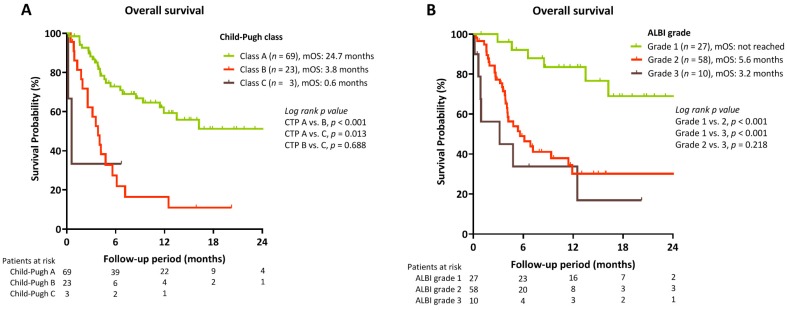
OS of HCC patients stratified by (**A**) Child–Pugh classification, and (**B**) Albumin-bilirubin (ALBI) grade.

**Table 1 cancers-12-00182-t001:** Characteristics and outcomes of hepatocellular carcinoma (HCC) patients treated with immune checkpoint inhibitors.

Characteristics	*n* = 95
Age, y	65.5 (57.2–72.9)
Sex (male), *n* (%)	73 (76.8)
HBsAg-positive, *n* (%)	62 (65.3)
Anti-HCV-positive, *n* (%)	21 (22.1)
Max. tumor size, cm	5.2 (2.3–8.8)
Tumor >50% liver volume, *n* (%)	30 (31.6)
Multiple tumors, *n* (%)	89 (93.7)
Extrahepatic metastasis, *n* (%)	48 (50.5)
Portal vein invasion, *n* (%)	51 (53.7)
AFP, ng/mL	609.7 (37.5–4832.3)
<10 ng/mL, *n* (%)	15 (15.8)
10–400 ng/mL, *n* (%)	27 (28.4)
≥400 ng/mL, *n* (%)	53 (55.8)
BCLC stage B/C, *n* (%)	20/75 (21.1/78.9)
Prothrombin time, INR	1.10 (1.05–1.23)
Platelet count, K/cumm	145 (102–218)
ALT, U/L	39 (25–61)
AST, U/L	57 (35–97)
Total bilirubin, mg/dL	1.03 (0.55–1.52)
Albumin, g/dL	3.6 (3.2–4.0)
Neutrophil-lymphocyte ratio	4.16 (2.89–6.85)
Presence of ascites, *n* (%)	37 (38.9)
Child–Pugh score	6 (5–7)
Child–Pugh class A/B/C, *n* (%)	69/23/3 (72.6/24.2/3.2)
ALBI grade 1/2/3, *n* (%)	27/58/10 (28.4/61.1/10.5)
First line systemic therapy, *n* (%)	39 (41.1)
Prior therapy to ICI, *n* (%)	
Surgical resection	35 (36.8)
RFA/PEIT/MWA	31/9/1 (32.6/9.5/1.1)
TACE/RT/TARE (Y-90)	55/23/5 (57.9/24.2/5.3)
Sorafenib	56 (58.9)
Nivolumab/Pembrolizumab, *n* (%)	92/3 (96.8/3.2)
Combined ICI with TKI, *n* (%)	13 (13.7)
Immune-related AEs	
Skin reactions/Pneumonitis/Hepatitis	6/4/3 (6.3/4.2/3.2)
Post PD treatment, *n* (%)	
TACE/RT/TARE (Y-90)	9/8/2 (9.5/8.4/2.1)
Regorafenib/Lenvatinib/Carbozantinib	8/16/2 (8.4/16.8/2.1)
Ramucirumab	4 (4.2)
Sorafenib/Traditional CT	7/6 (7.4/6.3)
Death	47 (49.5)

The data are expressed as median (interquartile range) unless marked with number (percentage) in behind. Abbreviations: AEs, adverse events; AFP, alpha fetoprotein; ALBI grade, albumin-bilirubin grade; ALT, alanine aminotransferase; AST, aspartate aminotransferase; BCLC stage, Barcelona-Clinic liver cancer stage; CI, confidence interval; CT, chemotherapy; HBsAg, hepatitis B surface antigen; HCV, hepatitis C; ICI, immune checkpoint inhibitor; INR, international normalized ratio; MWA, microwave ablation; PD, progressive disease; PEIT, percutaneous ethanol injection in tumor; RFA, radiofrequency ablation; RT, radiotherapy; TACE, transarterial chemoembolization; TARE (Y-90), transarterial radioembolization (Yttrium-90); TKI, tyrosine kinase inhibitors.

**Table 2 cancers-12-00182-t002:** Treatment response to immune checkpoint inhibitors.

Evaluable Response	All Patients (*n* = 95)	Child–Pugh A (*n* = 69)	Child–Pugh B (*n* = 23)	Child–Pugh C (*n* = 3)	Combination Treatment (*n* = 13)	Monotherapy (*n* = 82)
**Best Response, *n* (%)**						
Complete response	6 (6.7)	5 (7.5)	1 (5.0)	0	1 (7.7)	5 (6.5)
Partial response	16 (17.8)	13 (19.4)	3 (15.0)	0	5 (38.5)	11 (14.3)
Stable disease	11 (12.2)	10 (14.9)	1 (5.0)	0	1 (7.7)	10 (13.0)
Progressive disease	57 (63.3)	39 (58.2)	15 (75.0)	3 (100.0)	6 (46.2)	51 (66.2)
Non-assessable	5	2	3	0	0	5
Objective response rate	22 (24.4)	18 (26.9)	4 (20.0)	0	6 (46.2)	16 (20.8)
Disease control rate	33 (36.7)	28 (41.8)	5 (25.0)	0	7 (53.8)	26 (33.8)
**For Responders**						
Time to response (days)	63 (48–75)	64 (52–76)	52 (21–72)	–	57 (43–73)	63 (55–77)
Duration of response (months)	Not yet reached (16 ongoing)	Not yet reached (13 ongoing)	Not yet reached (three ongoing)	–	Not yet reached (five ongoing)	Not yet reached (11 ongoing)

**Table 3 cancers-12-00182-t003:** Factors associated with best objective response in 90 patients with evaluable responses.

Characteristics		Univariate Analysis			Multivariate Analysis		
		OR	95% CI	*p* Value	OR	95% CI	*p* Value
Age, y	>60 vs. ≤60	0.447	0.167–1.192	0.108			
Sex	Male vs. Female	0.691	0.228–2.092	0.514			
HBsAg-positive	Yes vs. No	1.651	0.573–4.756	0.353			
Anti-HCV-positive	Yes vs. No	0.722	0.213–2.446	0.601			
Tumor size, cm	>7 vs. ≤7	0.754	0.271–2.094	0.588			
Tumor number	Multiple vs. single	0.625	0.106–3.670	0.625			
Tumor shape	Infiltrative vs. nodular	2.250	0.813–6.227	0.118			
Tumor/Liver volume	>50% vs. ≤50%	0.900	0.308–2.633	0.847			
Portal vein invasion	Yes vs. No	1.131	0.431–2.969	0.802			
Main portal vein invasion	Yes vs. No PVI	1.046	0.278–3.932	0.947			
Portal branches invasion	Yes vs. No PVI	1.295	0.441–3.803	0.638			
Extrahepatic metastasis	Yes vs. No	0.580	0.219–1.537	0.273			
BCLC stage	Stage C vs. B	1.385	0.409–4.689	0.601			
AFP, ng/mL	>400 vs. ≤400	0.789	0.301–2.068	0.630			
AFP, ng/mL	<10 vs. ≤10	0.737	0.188–2.894	0.662			
NLR	>2.5 vs. ≤2.5	1.529	0.390–5.992	0.542			
Prothrombin time, INR	>1.2 vs. ≤1.2	1.211	0.422–3.470	0.722			
Platelet count	>100K vs. ≤100K	0.821	0.275–2.447	0.723			
ALT, U/L	> 40 vs. ≤40	0.294	0.097–0.888	0.030	0.384	0.109–1.349	0.135
AST, U/L	> 40 vs. ≤40	0.465	0.172–1.255	0.131			
Ascites	Yes vs. No	0.536	0.186–1.539	0.246			
Child–Pugh class	Class B, C vs. A	0.537	0.172–1.914	0.366			
ALBI grade	Grade 2,3 vs. 1	0.520	0.190–1.422	0.203			
Prior Sorafenib treatment	Yes vs. No	1.011	0.380–2.687	0.982			
Combined treatment *	Yes vs. No	3.813	1.083–13.419	0.037	2.522	0.572–11.111	0.222
AFP reduction at fourth week ^†^	Yes vs. No	7.437	2.545–21.735	<0.001	7.259	2.359–22.337	0.001
IO related AEs	Yes vs. No	0.916	0.228–3.678	0.901			

Abbreviations: AEs, adverse events; AFP, alpha fetoprotein; ALBI grade, albumin-bilirubin grade; ALT, alanine aminotransferase; AST, aspartate aminotransferase; BCLC stage, Barcelona-Clinic liver cancer stage; CI, confidence interval; HBV, hepatitis B; HCV, hepatitis C; INR, international normalized ratio; IO, immunotherapy; OR, odds ratio; NLR, neutrophil-lymphocyte ratio. * Combined treatment: combined immune checkpoint inhibitors with tyrosine kinase inhibitors, including sorafenib, lenvatinib, and regorafenib. ^†^ AFP reduction at fourth week: AFP reduced >10% from baseline serum level.

**Table 4 cancers-12-00182-t004:** Factors associated with overall survival in 95 patients treated with immune checkpoint inhibitors.

Characteristics		Univariate			Multivariate (Model 1) #			Multivariate (Model 2) #		
		HR	95% CI	*p*	HR	95% CI	*p*	HR	95% CI	*p*
Age, y	>60 vs. ≤60	1.252	0.676–2.318	0.476			NA			NA
Sex	Male vs. Female	0.632	0.337–1.186	0.153			NA			NA
HBsAg-positive	Yes vs. No	1.020	0.555–1.874	0.950			NA			NA
Anti-HCV-positive	Yes vs. No	1.393	0.729–2.661	0.315			NA			NA
Tumor size, cm	>7 vs. ≤7	2.450	1.362–4.409	0.003			NS			NS
Tumor number	Multiple vs. single	3.709	0.510–26.946	0.195			NA			NA
Tumor/Liver volume	>50% vs. ≤50%	2.425	1.323–4.444	0.004			NS			NS
Portal vein invasion	Yes vs. No	1.829	1.008–3.321	0.047			NS			NS
Extrahepatic metastasis	Yes vs. No	1.444	0.804–2.591	0.219			NA			NA
BCLC stage	Stage C vs. B	1.854	0.828–4.154	0.134			NA			NA
AFP, ng/mL	>400 vs. ≤400	2.039	1.102–3.773	0.023			NS			NS
AFP, ng/mL	<10 vs. ≤10	0.255	0.079–0.826	0.023			NS			NS
NLR	>2.5 vs. ≤2.5	1.010	0.467–2.185	0.981			NA			NA
Prothrombin time, INR	>1.2 vs. ≤1.2	1.585	0.842–2.983	0.154			NS			NS
Platelet count	>100K vs. ≤100K	0.928	0.479–1.799	0.825			NA			NA
ALT, U/L	>40 vs. ≤40	2.463	1.370–4.428	0.003			NS			NS
AST, U/L	>40 vs. ≤40	4.762	2.015–11.255	<0.001			NS			NS
Ascites	Yes vs. No	2.782	1.551–4.989	0.001			NA			NS
Child–Pugh class	Class A vs. B	0.260	0.143–0.472	<0.001	0.289	0.134–0.624	0.002			NA
ALBI grade	Grade1 vs. 2/3	0.189	0.079–0.453	<0.001			NA	0.220	0.084–0.576	0.002
Prior Sorafenib treatment	Yes vs. No	0.952	0.528–1.717	0.870			NA			NA
Combined treatment *	Yes vs. No	0.408	0.125–1.331	0.137			NS			NS
AFP reduction at fourth week ^†^	Yes vs. No	0.372	0.172–0.809	0.013	0.234	0.096–0.569	0.001	0.243	0.104–0.565	0.001
Immunotherapy related AEs	Yes vs. No	0.746	0.294–1.893	0.537			NA			NA

Abbreviations: ALBI grade, albumin-bilirubin grade; AEs, adverse events; AFP, alpha fetoprotein; AL(S)T, alanine(aspartate) aminotransferase; BCLC stage, AEs, adverse events; AFP, alpha fetoprotein; ALBI grade, albumin-bilirubin grade; ALT, alanine aminotransferase; AST, aspartate aminotransferase; BCLC stage, Barcelona-Clinic liver cancer stage; CI, confidence interval; HBsAg, hepatitis B surface antigen; HCV, hepatitis C; INR, international normalized ratio; NLR, neutrophil-lymphocyte ratio. * Combined treatment: combined immune checkpoint inhibitors with tyrosine kinase inhibitors, including sorafenib, lenvatinib, and regorafenib. ^†^ AFP reduction at fourth week: AFP reduced >10% from baseline serum level. # Model 1 enrolled significant parameters in univariate analysis into multivariate analysis, except ascites and ALBI grade. Model 2 enrolled significant parameters in univariate analysis into multivariate analysis, except Child-Pugh class.

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
