# Peer review of "Predictors of Response and Survival in Immune Checkpoint Inhibitor-Treated Unresectable Hepatocellular Carcinoma"

_cancers, 2020, doi:10.3390/cancers12010182_

Round 1
Reviewer 1 Report
I enjoyed reading this manuscript.However,I have some comments that may be taken in consideration.
1-The methodology part is almost missing in the abstract.
2-Some paragraphs in the introduction are lacking references.
ex1:Line 44 and 45
ex2:line 54 and 55
ex3:line 56 and 57
3-In the methods section.
4.3 PD-L1 expression analysis: A clarification needed why the authors selected 18 patients only? and images for IHC will be appreciated.
4.4 Biochemical and virological tests: under this title,no virological tests are mentioned!
Author Response
Response to Reviewer 1 Comments
Reviewer 1
I enjoyed reading this manuscript. However, I have some comments that may be taken in consideration.
Point 1: The methodology part is almost missing in the abstract.
Response 1: Thanks for your kind reminding. We have added methodology in the abstract. In order to meet the limitation of word counts, we also rewrote some sentences as follows:
“Ninety-five patients who received nivolumab or pembrolizumab for unresectable HCC were enrolled for analyses. Radiologic evaluation was based on RECIST v1.1. Factors associated with outcomes were analyzed. [presented in Line 24 – 27 in the revised manuscript]
Point 2: Some paragraphs in the introduction are lacking references, ex1:Line 44 and 45; ex2:line 54 and 55; and ex3:line 56 and 57.
Response 2: We have added the references in the following lines in our introduction and renewed the reference list accordingly.
The references we added and corrected were as follows:
Recently, several positive results from the phase 2/3 trials of first or second lines settings enable HCC patients to more treatment options [5-7].
Manipulation of immune checkpoints by targeted antibodies, such as anti-programmed cell death-1 (PD-1) antibody, has recently emerged as an effective anticancer strategy for many types of cancers including HCC [8]. Nivolumab and pembrolizumab, the anti-PD-1 antibodies, are FDA conditionally-approved immune checkpoint inhibitors (ICIs) for HCC as a second line treatment after sorafenib failure [5,6,9]. Based on the multi-cohort phase 1/2 trial CheckMate-040, phase 2 trial keynote-224 and phase 3 trial of Keynote-240 [5-7]
However, previous CheckMate-040 and Keynote-224 studies could not show significant association between PD-L1 expression level and tumor response in HCC [5,6]. As ICI treatment is expensive and has potential risk of immune-related adverse events, a baseline or early biomarker can help physicians to encourage suitable patients maintaining the treatment [2]
El-Khoueiry AB, Sangro B, Yau T, Crocenzi TS, Kudo M, Hsu C, Kim TY, Choo SP, Trojan J, Welling THR; et al. Nivolumab in patients with advanced hepatocellular carcinoma (CheckMate 040): an open-label, non-comparative, phase 1/2 dose escalation and expansion trial. Lancet. 2017,389(10088),2492-2502. Zhu AX, Finn RS, Edeline J, Cattan S, Ogasawara S, Palmer D, Verslype C, Zagonel V, Fartoux L, Vogel A; et al. Pembrolizumab in patients with advanced hepatocellular carcinoma previously treated with sorafenib (KEYNOTE-224): a non-randomised, open-label phase 2 trial. Lancet Oncol. 2018,19(7),940-952. Finn RS, Ryoo BY, Merle P, Kudo M, Bouattour M, Lim HY, Breder V, Edeline J, Chao Y, Ogasawara S; et al. Results of KEYNOTE-240: phase 3 study of pembrolizumab (Pembro) vs best supportive care (BSC) for second line therapy in advanced hepatocellular carcinoma (HCC). J Clin Oncol. 2019,37, no. 15 suppl,4004. Topalian SL, Hodi FS, Brahmer JR, Gettinger SN, Smith DC, McDermott DF, Powderly JD, Carvajal RD, Sosman JA, Atkins MB; et al. Safety, activity, and immune correlates of anti-PD-1 antibody in cancer. N Engl J Med. 2012,366(26),2443-2454. Finkelmeier F, Waidmann O, Trojan J. Nivolumab for the treatment of hepatocellular carcinoma. Expert Rev Anticancer Ther. 2018,18(12),1169-1175.Point 3: In the methods section.
4.3 PD-L1 expression analysis: A clarification needed why the authors selected 18 patients only? and images for IHC will be appreciated.
Response 3: Thanks for your important question. As not all the patients had archival tissues or received tumor biopsy before ICI treatment; therefore, only 18 patients had liver tissue samples for PD-L1 analysis.
Regarding the images for IHC, we have official reports from Taipei Institute of Pathology, Taiwan. Unfortunately, the images are not available, as the institute did not routinely keep the images forever.
Point 4: 4.4 Biochemical and virological tests: under this title, no virological tests are mentioned!.
Response 4: Thanks for your recommendation. We have deleted “virological” from this subtitle in our materials and methods.
The title we rewrote was as follows:
“4.4. Biochemical and virological tests” was rewritten as “4.4. Biochemical tests”
[presented in Line 252 in the revised manuscript]
Reviewer 2 Report
Lee and colleagues investigated the potential predictors of response and factors associated with OS for immune checkpoint inhibitor (ICI) treatment (with nivolumab or pembrolizumab) in 95 patients with unresectable HCC. Early AFP >10% reduction within 4 weeks was the only independent predictor of best objective response (OR 7.259, p=0.001). In addition, the early AFP reduction and ALBI grade or Child-Pugh class were independent factors associated with OS. This article is well written and informative for the Journal readers, but some specific points should be addressed.
Major point
The included patients were a bit heterogenous. For example, 13/95 patients received combination therapy with ICIs and tyrosine kinase inhibitors (Table 1); the combination treatment had significantly higher ORR than monotherapy (46.2% vs 20.8%, p=0.049). Was the results unchanged, when 13 patients with combination therapy were excluded?
Minor point
In the Discussion section, the authors discussed the ongoing clinical trials of combination therapy for advance-staged HCC (line 175). Among them, the efficacy and safety of atezolizumab plus bevacizumab have recently been released, probably after the authors submitted this paper. The authors had better provide more information regarding this combination, as the most promising example. Table 3 has a complete list of abbreviations in the footnote, but table 1 has no and table 4 has incomplete one. In the Tables, are data expressed as number (percentage) or median (IQR)?Author Response
Response to Reviewer 2 Comments
Reviewer 2
Lee and colleagues investigated the potential predictors of response and factors associated with OS for immune checkpoint inhibitor (ICI) treatment (with nivolumab or pembrolizumab) in 95 patients with unresectable HCC. Early AFP >10% reduction within 4 weeks was the only independent predictor of best objective response (OR 7.259, p=0.001). In addition, the early AFP reduction and ALBI grade or Child-Pugh class were independent factors associated with OS. This article is well written and informative for the Journal readers, but some specific points should be addressed.
Major point: The included patients were a bit heterogeneous. For example, 13/95 patients received combination therapy with ICIs and tyrosine kinase inhibitors (Table 1); the combination treatment had significantly higher ORR than monotherapy (46.2% vs 20.8%, p=0.049). Was the results unchanged, when 13 patients with combination therapy were excluded?
Response to major point:
Thanks for your important question. We had tried to exclude the patients receiving combination therapy for analysis, the results are still unchanged.
Among the 82 patients who received ICI monotherapy with baseline AFP ≥10 ng/ml, significantly higher objective response rate (55.6% vs. 7.0%, p<0.001) and disease control rate (77.8% vs. 11.6%, p<0.001) were still observed in those with early AFP reduction than those without. Besides, early AFP response was still the only independent predictor of best objective response to ICI treatment (odds ratio: 6.818, p=0.002).
In addition, patients with early AFP reduction >10% also had significantly better OS than those without AFP response (median OS: 24.7 vs. 6.1 months, p=0.005). Child-Pugh A vs B/C (median OS: 16.2 vs. 3.8 and 0.6 months, p<0.001), and ALBI grade 1 vs 2/3 (median OS: not yet reached, vs 4.8 and 3.2 months; p<0.001) (Figure 4B) were still associated with OS. In multivariate analysis, early AFP reduction (hazard ratio [HR]: 0.205, p=0.001) and Child-Pugh A (HR: 0.253, p=0.001) were the independent predictors to better OS in patients received ICI monotherapy (model 1). In the model 2, early AFP response (HR: 0.167, p<0.001) and ALBI grade 1 (HR: 0.133, p<0.001) were also independent good survival predictors.
Minor point 1: In the Discussion section, the authors discussed the ongoing clinical trials of combination therapy for advance-staged HCC (line 175). Among them, the efficacy and safety of atezolizumab plus bevacizumab have recently been released, probably after the authors submitted this paper. The authors had better provide more information regarding this combination, as the most promising example..
Response to Minor point 1: Thanks very much for your recommendation. We have added more description about the efficacy and safety of atezolizumab plus bevacizumab which was recently declared in ESMO Asia congress 2019.
The description we added in Discussion were as follows:
As recently reported in ESMO Asia 2019, the phase 3 IMbrave 150 has demonstrated significant improvements with atezolizumab and bevacizumab over sorafenib in OS and RFS for unresectable HCC (ORR 27%, DCR 74% by RECIST 1.1).
[presented in Line 192 – 194 in the revised manuscript]
Minor point 2: Table 3 has a complete list of abbreviations in the footnote, but table 1 has no and table 4 has incomplete one. In the Tables, are data expressed as number (percentage) or median (IQR)?
Response to Minor point 2: Thanks for your kind reminding. We have added on the footnotes of table 1 and 4. We also mentioned the expression format of our data in the footnote.
The footnotes of table 1 were as follows:
The data were expressed as median (interquatile range) unless marked with number (percentage) in behind.
Abbreviations: AEs, adverse events; AFP, alpha fetoprotein; ALBI grade, albumin-bilirubin grade; ALT, alanine aminotransferase; AST, aspartate aminotransferase; BCLC stage, Barcelona-Clinic liver cancer stage; CI, confidence interval; CT, chemotherapy; HBsAg, hepatitis B surface antigen; HCV, hepatitis C; ICI, immune checkpoint inhibitor; INR, international normalized ratio; MWA, microwave ablation; PD, progressive disease; PEIT, percuatenous ethanol injection in tumor; RFA, radiofrequency abalation; RT, radiotherapy; TACE, transarterial chemoembolization; TARE (Y-90), transarterial radioembolization (Yttrium-90); TKI, tyrosine kinase inhibitors.
[presented in Line 73 – 82 in the revised manuscript.]
The footnotes of table 4 were as follows:
Abbreviations: ALBI grade, albumin-bilirubin grade; AEs, adverse events; AFP, alpha fetoprotein; AL(S)T, alanine(aspartate) aminotransferase; BCLC stage, AEs, adverse events; AFP, alpha fetoprotein; ALBI grade, albumin-bilirubin grade; ALT, alanine aminotransferase; AST, aspartate aminotransferase; BCLC stage, Barcelona-Clinic liver cancer stage; CI, confidence interval; HBsAg, hepatitis B surface antigen; HCV, hepatitis C; INR, international normalized ratio; NLR, neutrophil-lymphocyte ratio. *Combined treatment: combined immune checkpoint inhibitors with tyrosine kinase inhibitors, including sorafenib, lenvatinib, and regorafenib.
†AFP reduction at 4th week: AFP reduced > 10% from baseline serum level.
[presented in Line 149 – 153 in the revised manuscript.]